# Epigenetic Approaches to Overcome Fluoropyrimidines Resistance in Solid Tumors

**DOI:** 10.3390/cancers14030695

**Published:** 2022-01-29

**Authors:** Laura Grumetti, Rita Lombardi, Federica Iannelli, Biagio Pucci, Antonio Avallone, Elena Di Gennaro, Alfredo Budillon

**Affiliations:** 1Experimetnal Pharmacology Unit-Laboratory of Naples and Mercogliano (AV), Istituto Nazionale Tumori IRCCS “Fondazione G. Pascale”, 80131 Naples, Italy; laura.grumetti@istitutotumori.na.it (L.G.); r.lombardi@istitutotumori.na.it (R.L.); f.iannelli@istitutotumori.na.it (F.I.); b.pucci@istitutotumori.na.it (B.P.); 2Experimental Clinical Abdominal Oncology Unit, Istituto Nazionale Tumori di Napoli IRCCS “Fondazione Pascale”, 80131 Naples, Italy; a.avallone@istitutotumori.na.it

**Keywords:** HDAC inhibitors, fluoropyrimidines, drug resistance

## Abstract

**Simple Summary:**

Fluoropyrimidines represent the backbone of many combination chemotherapy regimens for the treatment of solid cancers but are still associated with toxicity and mechanisms of resistance. In this review, we focused on the epigenetic modifiers histone deacetylase inhibitors (HDACis) and on their ability to regulate specific genes and proteins involved in the fluoropyrimidine metabolism and resistance mechanisms. We presented emerging preclinical and clinical studies, highlighting the mechanisms by which HDACis can prevent/overcome the resistance and/or enhance the therapeutic efficacy of fluoropyrimidines, potentially reducing their toxicity, and ultimately improving the overall survival of cancer patients.

**Abstract:**

Although fluoropyrimidines were introduced as anticancer agents over 60 years ago, they are still the backbone of many combination chemotherapy regimens for the treatment of solid cancers. Like other chemotherapeutic agents, the therapeutic efficacy of fluoropyrimidines can be affected by drug resistance and severe toxicities; thus, novel therapeutic approaches are required to potentiate their efficacy and overcome drug resistance. In the last 20 years, the deregulation of epigenetic mechanisms has been shown to contribute to cancer hallmarks. Histone modifications play an important role in directing the transcriptional machinery and therefore represent interesting druggable targets. In this review, we focused on histone deacetylase inhibitors (HDACis) that can increase antitumor efficacy and overcome resistance to fluoropyrimidines by targeting specific genes or proteins. Our preclinical data showed a strong synergistic interaction between HDACi and fluoropyrimidines in different cancer models, but the clinical studies did not seem to confirm these observations. Most likely, the introduction of increasingly complex preclinical models, both in vitro and in vivo, cannot recapitulate human complexity; however, our analysis of clinical studies revealed that most of them were designed without a mechanistic approach and, importantly, without careful patient selection.

## 1. Introduction

Cancer chemotherapy is one of the most established and effective treatments for almost all types of cancer. However, this approach does not discriminate between rapidly dividing nonmalignant cells and cancer cells, leading to nontumor-associated effects that produce elevated toxicity. In addition, during cancer progression, tumors become highly heterogeneous and create a mixed population of cells characterized by different molecular features and diverse responsivity to therapy. This heterogeneity is the key factor responsible for the development of resistant tumor phenotypes, which are promoted by the selective pressure of chemotherapy administration that limits the effectiveness and safety of treatment. A deeper understanding of these complex phenomena is essential to design novel therapeutic approaches that address the challenge of chemotherapy resistance [1]

Fluoropyrimidines, such as 5-fluorouracil (5-FU) and the prodrug capecitabine, are the backbone of many combination chemotherapy regimens for the treatment of solid cancers, including gastrointestinal, breast, pancreas and head and neck cancers. Despite their clinical benefits, fluoropyrimidines are associated with both toxicity and mechanisms of resistance that could affect therapeutic efficacy. Indeed, current treatment strategies are ineffective in many patients; thus, novel therapeutic approaches are required to potentiate the efficacy of fluoropyrimidines and overcome mechanisms of resistance [2].

Epigenetic characteristics allow individual cell/tissue types to maintain their unique identity and to differentially express genes suitable for their biological function. DNA methylation and covalent histone modifications are the two major hallmarks of epigenetic regulation and alteration in the epigenetic networks of cancers [3]. The epigenetic elements involved in different modification patterns can be divided into three roles: “writers,” “erasers” and “readers”. The “writers” (DNA methyltransferases, histone acetyl transferase and histone methyltransferases) and “erasers” (DNA-demethylating enzymes, histone deacetylases, and histone-demethylating enzymes) refer to enzymes that transfer/remove chemical groups to/from DNA or histones, respectively. Importantly, all three families of epigenetic proteins (readers, writers, and erasers) involved in histone modifications play an important role in directing the transcriptional machinery and represent interesting druggable targets. Histone modifications have been investigated in many disease areas, including solid tumors, hematological malignancies, and even many inflammatory diseases (such as viral infection, diabetes and inflammatory lung diseases). In the past two decades, the acetylation of histone molecules has attracted increasing attention; these molecules are involved in the complex regulation of genome properties, including transcription and DNA repair [4,5].

In this review, we will discuss how epigenetic approaches, and histone deacetylase inhibitors (HDACis), can play a role in priming activity and overcoming resistance to fluoropyrimidine-based therapy.

## 2. HDAC Inhibitors

Histone acetylation is tightly controlled by a balance between the opposing activities of histone acetyltransferases (HATs) and histone deacetylases (HDACs). HATs modify core histone tails by post-translational acetylation of the amino-terminal ε-group of lysines on H3 and H4 histone tails, thereby creating an appropriate ‘histone code’ for chromatin modification and enhancing the DNA accessibility of transcription factors. HDACs act as gene-silencing mediators and repress the transcription process by deacetylating the same lysine residues. Importantly, both HDACs and HATs are expressed not only in the nucleus but also in the cytoplasm, thereby regulating the acetylation of different nonhistone proteins [6]. Disruptions to the balance between HAT and HDAC activity can result in the aberrant expression of genes that ultimately leads to the instability of chromatic structures and epigenetic diseases, including solid tumors and hematological malignancies [3,7].

Deacetylases can be divided into two families based on the presence or absence of a conserved deacetylase domain and their dependence on specific cofactors: the zinc-dependent histone deacetylase (HDAC) family and the sirtuins protein family. To date, 11 mammalian zinc-dependent amidohydrolase HDACs have been reported, which are subdivided into class I (HDAC1, 2, 3, and 8), class II (HDAC4, 5, 6, 7, 9, and 10), and class IV (HDAC11). According to the composition of their domains, the class II enzymes are further divided into two subclasses, IIa and IIb. The class III deacetylases or sirtuins are a distinct group of enzymes that require nicotinamide adenine dinucleotide (NAD) as a cofactor for their catalytic function and will not be further discussed in the present review article [8] (Figure 1).

Although identification of the substrate specificity and the biological function of individual HDACs still requires more comprehensive investigation, it is well known that HDACs play crucial roles in cancer progression, apoptosis, cell cycle control, angiogenesis, and cell invasion [3] (Figure 2), providing a rationale for targeting HDACs in cancer therapy via HDAC inhibitors (HDACis). Furthermore, cancer cells are more sensitive to HDACi-induced apoptosis than normal cells [9], supporting the therapeutic potential of HDACis [3].

HDACis can be divided into four groups according to their chemical structure: aliphatic fatty acids, hydroximic acids, benzamides, and cyclic peptides. Moreover, based on their specificity, HDACis can be divided into three additional groups: (1) nonselective or pan-HDACis, such as vorinostat, belinostat, and panobinostat; (2) selective HDACis, such as class I HDACis (romidepsin and entinostat) and HDAC6 inhibitor (ricolinostat) [10]; and (3) multipharmacological HDACis, which target both an HDAC and another target, such as CUDC-101 (a multiple HDAC/EGFR/HER2 inhibitor [11]) and CUDC-907 (a dual HDAC/PI3K inhibitor [12]) (Figure 1).

Currently, numerous HDACis are in clinical development as anticancer drugs, and three of them (vorinostat, romidepsin and belinostat) have been approved for the treatment of cutaneous T-cell lymphoma by the US FDA; panobinostat has been approved in combination therapy to treat recurrent multiple myeloma [13,14,15,16,17].

Although the clinical efficacy of HDACis in monotherapy for solid tumors is limited, HDACis acting as sensitizers and modulators of the entire gene pattern could act synergistically with many treatments, including standard chemotherapy, targeted therapy, DNA repair pathway drugs, radiotherapy, and immune-based therapies, priming their activity or overcoming the resistance often associated with antitumor approaches [5,18]. Consequently, a variety of combinatorial therapeutic strategies have attracted increasing attention towards the possibility of translating preclinical data into clinical studies [8]. An important aspect to consider in the design of combinational preclinical and clinical studies is deep knowledge of the molecular targets as well as the mechanism of action of the drugs, which can be used to define the optimal dosage and schedule of administration with the aim of maximizing efficacy and preventing toxicity [18].

Our group has intensely researched HDACis, publishing studies on the mechanism underlying the synergistic effect of HDACis and chemotherapeutics [19,20,21,22] or anti-EGFR agents [23,24,25,26], as well as immunotherapy [27], and launching ongoing clinical trials with such combinatory approaches [28,29].

## 3. Fluoropyrimidines

The antimetabolite 5-fluorouracil (5-FU) was introduced as an anticancer agent over 60 years ago and still composes the backbone of treatment for different types of cancers, along with other fluoropyrimidines, such as the oral prodrug capecitabine.

Two competing routes are responsible for 5-FU metabolism: the anabolic route, which transforms 5-FU into active metabolites, and the catabolic route, which inactivates and excretes 5-FU.

The mechanism of 5-FU cytotoxicity has been ascribed to the misincorporation of fluoronucleotides into the RNA and DNA of its active metabolites, 5-fluorouridine triphosphate (5FUTP) and 5-fluorodeoxyuridine triphosphate (5FdUTP), and to the inhibition of thymidylate synthase (TS) by 5-fluorodeoxyuridine monophosphate (5FdUMP), leading to the disruption of the intracellular deoxynucleotide pools required for DNA replication (Figure 3).

Capecitabine is the oral prodrug that is converted into 5-FU. Three intracellular metabolites are responsible for the antineoplastic effect of these drugs. In brief, FUTP is incorporated into RNA and interferes with normal RNA processing and function also contributing to toxicity. FdUTP is incorporated into DNA, leading to DNA damage and cell death. FdUMP inhibits thymidylate synthase, the enzyme that catalyzes the transformation of deoxyuridine monophosphate (dUMP) to deoxythymidine monophosphate (dTMP). Inhibition of thymidylate synthase by FdUMP leads to accumulation of deoxyuridine triphosphate (dUTP) and depletion of deoxythymidine triphosphate (dTTP). This imbalance has deleterious consequences for DNA synthesis and repair, leading to cell death.

TS plays a key role in DNA synthesis, catalyzing the conversion of deoxyuridine monophosphate (dUMP) into thymidylate (dTMP), with the methyl donor 5,10-methylene tetrahydrofolate (CH2THF), representing the sole intracellular source of dTMP. The inhibition of dTMP synthesis by 5FdUMP is due to the formation of a stable ternary complex between TS, CH2THF, and dUMP, which prevents the transfer of a methyl group to carbon 5 of dUMP to form dTMP [30].

This inhibition of dTMP synthesis results in the subsequent depletion of deoxythymidine triphosphate (dTTP) and an imbalance in the other deoxynucleotides (dATP, dGTP and dCTP), which affects DNA synthesis and repair and causes lethal DNA damage. Moreover, TS inhibition results in the accumulation of dUMP, leading to increased levels of deoxyuridine triphosphate (dUTP).

Furthermore, 5-FU can also be incorporated into DNA through conversion into 5-fluoro-2-deoxyuridine (FdUR) by thymidine phosphorylase (TP) and then into fluorodeoxyuridine monophosphate (FdUMP) by thymidine kinase (TK). Through specific enzymatic reactions, FdUMP is converted into FdUTP, which can be misincorporated into DNA in a similar manner to dUTP [2,30]. Moreover, 5-FU can also be converted into the active metabolite fluorouridine triphosphate (FUTP), which can be integrated into RNA [2,30] (Figure 3).

More than 80% of administered 5-FU is degraded by dihydropyrimidine dehydrogenase (DPD) in the liver, where this enzyme is abundantly expressed [30,31]. DPD, the rate-limiting enzyme of catabolism, reduces 5-FU to 5,6-dihydro-5-fluorouracil (DHFU), and it is subsequently excreted via the kidneys [30]. Interestingly, DPD activity has been reported to be influenced by genetic variation (interpatient variability) as well as circadian rhythms (intrapatient variability). Accordingly, 5-FU bioavailability may be influenced both by DPD expression levels and by the drug administration modality (bolus, infusion or oral prodrug). When 5-FU is administered by infusion, nearly 20% of the dose is directly excreted in the urine [2].

To reduce 5-FU toxicity, extend its duration of action, and increase its tumor selectivity, molecules that act as prodrugs of 5-FU were developed. Due to their ease of administration, tegafur and capecitabine are the main prodrugs administered in daily clinical practice. Both drugs are administered orally and are designed to be absorbed through the gastrointestinal mucosa and subsequently enzymatically converted into 5-FU in the liver or within the tumor itself. Tegafur is metabolized by cytochrome P450, mainly in the liver, and converted into 5-FU, but it is simultaneously catabolized and degraded by DPD. Indeed, to improve the therapeutic index of tegafur, other molecules were designed to block DPD-mediated degradation. First, tegafur-uracil, which comprises tegafur and uracil in molar proportions of 1:4, was designed so that uracil could compete with 5-FU for DPD activity after incorporation into RNA and thus potentiates the effect of tegafur. The second, S-1, was developed to enhance the effect of tegafur-uracil and reduce side effects and consists of tegafur, gimeracil, and oteracil in a molar ratio of 1:0.4:1. Gimeracil inhibits the DPD enzyme more potently (200-fold) than uracil, while oteracil enhances the antitumor effect and reduces gastrointestinal toxicity through its inhibition of 5-FU phosphorylation and its distribution at high concentrations in the gastrointestinal tract [2,32].

Capecitabine is the other prodrug of 5-FU largely used in clinical practice and was developed as a 5′-DFUR prodrug, to prevent the metabolic transformation of 5′-DFUR by TP. Indeed, capecitabine is converted into 5-FU through a series of sequential steps. It is first absorbed through the gastrointestinal wall in an intact form and, subsequently, is converted to 5′-DFUR by carboxylesterase (CE) and cytidine deaminase (CDA) in the liver. 5′-DFUR is then transformed to 5-FU by TP and/or UP. TP is the key enzyme that converts prodrugs in active 5-FU; thus, its expression may be correlated with the efficacy of 5-FU-based chemotherapy [2,32] (Figure 3).

## 4. Mechanisms of Resistance to Fluoropyrimidines

Fluoropyrimidine resistance is principally controlled by the three major enzymes involved in 5-FU metabolism and described above (TS, TP and DPD); however, other mechanisms not directly related to the metabolism of fluoropyrimidines have been identified, such as cancer stemness, angiogenesis and DNA repair [33]. The critical role of TS expression in primary 5-FU resistance was established long ago, even before the oncogene-like activity of TS was reported in 2004 [34]. Indeed, it is currently widely accepted that elevated TS expression in cancer is the major molecular mechanism of 5-FU resistance. Several clinical trials and meta-analyses have demonstrated that, independent of cancer type, patients with low TS expression in tumor tissue have longer overall survival and higher sensitivity to 5-FU-based chemotherapy than those with higher TS expression levels [35,36,37,38,39].

TS protein expression is regulated in a complex way at both the transcriptional and translational levels [40]. Several polymorphisms have been reported in the TS gene (TYMS) promoter, which interfere with the regulation of TYMS expression and thus affect 5-FU sensitivity [41,42,43,44]. Conversely, it has been reported that the increased expression of TS may be the consequence of the overexpression of the transcription factor E2F1 [45]. In support of this theory, Kasahara M et al. reported that TS expression correlated closely with transcription factor E2F1 expression in 23 colon cancer patient samples [46]. Interestingly, heat shock protein 90 (HSP90), a chaperone protein that regulates the stability and trafficking of several client proteins involved in cell proliferation by regulating E2F levels and gene transcription [47], is implicated in the transcriptional overexpression of TYMS. Moreover, the activation of the HSP90–Src signaling pathway was identified as a novel mechanism for acquired resistance to 5-FU in CRC cell lines [48]. Furthermore, the downregulation of TS expression upon HSP90 inhibition sensitized colorectal cancer cell lines to the effect of 5-FU-based chemotherapy [49].

TS protein expression is also regulated by a negative-feedback mechanism in which TS binds its own mRNA, thus inhibiting TS protein translation [30,50,51]. This negative-feedback mechanism can be inhibited by the exposure of cancer cells to 5-FU or other TS inhibitors, and results in the consequent increase in TS expression [50,51]. Interestingly, the TS protein, acting as an RNA binding protein, also decreases the expression of genes involved in the regulation of proliferative and survival pathways, such as c-myc and p53. Conversely, TS transcription appears to be inhibited by p53, but this relationship may be altered by other mutations affecting p53 [41,52,53].

miR-203 and miR-330 that target TYMS have been reported to reduce its protein level, enhancing the antitumor activity of 5-FU. [54,55]. Similarly, Li et al. identified the prognostic value of miR-218 in CRC patients, reporting that high miR-218 expression promoted apoptosis and sensitized CRC cells to 5-FU treatments by suppressing TS and BIRC5 expression [56].

Another miRNA, miR-375-3p, which targets the oncogenic transcription factors YAP1 and SP1, suppressed tumorigenesis and partially reversed chemoresistance in colorectal cancers [57]. More recently, it has been shown that miR-375-3p, by targeting TS in human CRC cell lines and tissues, enhanced chemosensitivity to 5-FU, inducing apoptosis and cell cycle arrest and inhibiting cell growth, migration, and invasion in vitro [58].

The role of thymidine phosphorylase (TP) in the clinical response to fluoropyrimidine-based chemotherapy is complex. Indeed, as described previously, TP is the enzyme responsible for the conversion of the prodrug capecitabine (5′DFUR) into 5-FU. Moreover, TP is also the critical enzyme for converting 5-FU into the metabolites responsible for TS inhibition. Notably, TP has strong sequence homology with proangiogenic platelet-derived endothelial cell growth factor (PD-ECGF), thus contributing to angiogenesis, tumor progression and metastasis in cancer cells [31]. These observations underline the dual and controversial role of TP in cancer development and treatment [59].

Thus, if patients affected by different solid tumors, including pancreatic, colon, gastric and renal cancer tumors, show high TP expression, they have poorer prognoses than those with low TP expression [60,61,62,63]; patients with high levels of intratumoral TP expression are ideal candidates for capecitabine-based chemotherapy [64]. Regardless, higher levels of TP in tumor cells compared with normal tissues can explain the correlation with the efficacy of 5-FU-based chemotherapy in a wide range of solid tumors [65]. Interestingly, Meropol et al. showed that TP expression, measured by IHC, was associated with improved response rates, time to progression and overall survival in metastatic colorectal cancer patients treated first-line with capecitabine plus irinotecan [66]. However, in colorectal cancers, in which fluoropyrimidine-based regimens compose the backbone of treatment, a definitive conclusion between the levels of TP and survival has not been drawn [67,68,69,70].

To date, the mechanisms that regulate TP expression have not been completely defined. The promoter region of the TP gene (TYMP) is characterized by high G-C content and seven binding sites for the transcription factor SP1. The activation of SP1 by different factors, including inflammatory cytokines and tumor necrosis factor α (TNFα), plays an important role in TP regulation. Chemotherapy agents such as docetaxel, paclitaxel, cyclophosphamide, oxaliplatin and radiotherapy can increase TP levels through this mechanism, thus providing the rationale for combining antitumor therapeutic approaches with fluoropyrimidines [71,72].

As described previously, DPD, which is widely expressed in various cancers, including colorectal [73], gastric [74], lung [75], and oral [76] tumors, as well as in healthy tissues, such as in liver and peripheral blood mononuclear cells (PBMCs) [77], plays a crucial role in 5-FU sensitivity, as it is the main enzyme responsible for 5-FU catabolism (converting approximately 85% of administered 5-FU). The levels of DPD have been associated with lower response (high levels) or severe and life-threatening toxicity (low levels) to 5-FU [77]. Thus, DPYD gene regulation is important in the determination of enzyme activity, as it plays a crucial role in the clinical management of 5-FU [77]. The two transcription factors SP1 and SP3 play an important role in the transcriptional regulation of the DPYD gene. SP1 is a strong activator of constitutive expression of DPYD; SP3 is a weak activator but, when working together with SP1, acts as a negative regulator of the DPYD gene. Tumors with a high proliferation rate have less phosphorylation/activity of SP1, thus reducing DPYD gene expression [77].

Notably, the methylation of CpG sites in the DPYD promoter region is associated with downregulation of DPD activity [78], and these CpG island contain SP1 protein binding sites [79].

Zhang and colleagues showed that combining the demethylating agent AzaC and the HDACi trichostatin A increased DPYD expression [80], suggesting that histone deacetylation might have a role in silencing the DPYD gene when DNA methylation levels are low [81].

The expression and activity of DPD can also be affected by some polymorphisms. More than 13 sequence variants result in the dysfunction of DPD protein associated with DPD deficiency and an increase in 5-FU toxicity. The DPYD*2A variant results in a complete loss of DPD function and plays a major role in fluoropyrimidine-related adverse events. Approximately 2% of Caucasians of European descent possess this allele mutation; luckily, most patients are heterozygous and can be treated with reduced 5-FU dosing. However, although very rare (~1:1000), a complete deficiency of DPD expression can be lethal in homozygous patients treated with fluoropyrimidine chemotherapy [82]. On 30 April 2020, the European Society for Medical Oncology (ESMO) issued guidelines that recommended genetic testing of DPYD before starting treatment, an approach not yet followed by oncology societies in the United States [83].

DPD expression may also be regulated at the post-transcriptional level [84,85]. Preclinical data reported for a colorectal cancer cell line (SW480) demonstrated that miR-494, by interacting with the 3′UTR of the DPYD gene, negatively regulated endogenous DPYD expression [86]. Similarly, Offer et al. demonstrated that DPYD is a direct downstream target of miR-27a and miR-27b and that the overexpression of these two miRNAs may induce repression of DPD and increase sensitivity to 5-FU [87].

Recently, it was demonstrated that DNA damage responses, particularly the base excision repair (BER) and mismatch repair (MMR) pathways, are relevant for the response and outcome of 5-FU-treated patients [88]. It is well known that the DNA MMR system, composed of MLH1, MSH2, MSH3, MSH6 and PMS2 proteins, is responsible for maintaining genomic stability and DNA repair [89], and genetic or epigenetic events can result in nonfunctional proteins (deficient MMR, dMMR) causing a microsatellite instability (MSI) phenotype in several tumor types [90]. In CRC patients, MSI determination has emerged as a valuable tool to predetermine patients’ eventual responses to adjuvant 5-FU and categorize patients into prognostic subgroups [91]. Currently, dMMR CRC tumors are well known to have better clinical outcomes than proficient MMR (pMMR) tumors [33], partially due to a high mutational burden accompanied by abundant mutation-derived neoantigens that attract tumor infiltrating lymphocytes (TILs) in dMMR tumors [92].

The enzymes required for BER in the basic reaction step include uracil-DNA-glycosylases (UNGs), which catalyze the excision of uracil nucleobases from DNA due to misincorporation or spontaneous cytosine deamination [93]. Five human UNG isoforms have been identified; of these, UNG2 is the main enzyme and the quantitatively dominant form in proliferating cells. Interestingly, when UNG2 removes uracil, it causes other mutations in the immunoglobulin loci, responsible for somatic hypermutation (SHM), which increases immunoglobulin diversity. Moreover, UNG2 is also involved in the innate immune response against retroviral infections [94]. As UNG2 removes uracil from DNA, it can also remove 5-FU, thus mediating the 5-FU sensitivity of tumor cells. Indeed, UNG2 depletion leads to DNA fragmentation and the accumulation of uracil and/or 5-FU at replication forks, enhancing the cytotoxicity of 5-FU [95,96].

Additional mechanisms of fluoropyrimidine resistance in cancer cells include several molecular and cellular processes, such as the cell cycle, apoptosis, autophagy, oxidative stress, drug efflux pumps, and cancer stem cell (CSC) or epithelial-to-mesenchymal transition (EMT) pathways [97].

For example, the inhibition of the p38 MAPK signaling pathway in 5-FU-resistant cells mediates an autophagic response associated with the inhibition of p53-dependent apoptosis [98]. Interestingly, 5-FU-resistant cells are characterized by an increased ability to form spheres and colonies, migrate, and invade, typical features of cancer stem cells (CSCs), confirmed by the upregulation of stem cell markers, including NOTCH1, CD44, ALDHA1, Oct4, SOX2, and Nanog [99]. 5-FU resistance in CSCs is due to the acquisition of a quiescent state, a metabolic switch, aberrant activation of different growth signaling pathways and resistance to DNA damage [33,100,101,102].

Furthermore, it is well known that CSCs are able to develop cellular adaptive responses to reactive oxygen species (ROS) induced by anticancer agents, including 5-FU. 5-FU has been reported to induce the activation and nuclear translocation of Nrf2, resulting in the upregulation of antioxidant enzymes and in 5-FU resistance [97].

In addition, Touil and colleagues reported that in quiescent CRC CSCs, the c-Yes/YAP axis represents another mechanism of 5-FU resistance. The YES1 gene, whose chromosomal location is close to the TYMS gene is amplified in 5-FU-resistant CSCs, and, after 5-FU-based neoadjuvant chemotherapy, the transcript levels of both YES and YAP are higher in liver metastases of patients with CRC and positively correlate with CRC relapse and reduced patient survival [101].

EMT contributes to the emergence of CSCs, causing an increase in both metastasis and drug resistance. Indeed, resistance to 5-FU in CRC cells is associated with the repression of GDF15, a member of the TGFβ/bone morphogenetic protein superfamily involved in the regulation of EMT [103]. Romano et al. showed that 5-FU treatment of CRC models upregulated the TGF-β pathway through the activation of SMAD3 and the transcription of specific genes, such as ACVRL1, FN1 and TGFB1, and that drug sensitivity can be restored by specific inhibition of TGF-β signaling [104]. Similarly, the suppression of the well-known oncogene TWIST1 (which is induced by TGF-β treatment) sensitizes CRC cell lines to 5-FU-induced apoptosis [105]. However, the role of TGF-β signaling in 5-FU resistance is controversial since TGF-β signaling-deficient mice were recently found to have a specific gut microbiome signature associated with 5-FU resistance [106].

Overexpression of the ATP-binding cassette (ABC) transporter on the membrane of cancer cells is a broad mechanism of cancer cells’ resistance to anticancer drugs as it mediates ATP-dependent transport and efflux of anticancer agents out of cells [107,108].

Although fluoropyrimidines are not substrates of ABC transporters, Xie and colleagues reported that 5-FU treatment induces upregulation of the transcription factor FOXM1, which in turn upregulates the transporter MRP7/ABCC10, and that the inhibition of FOXM1 and/or ABCC10 was able to reverse 5-FU resistance [109].

p53 plays an important role in anticancer drug sensitivity, and the gain of function conferred by certain p53 mutants has been linked to fluoropyrimidine chemoresistance [110,111]. Moreover, several clinical studies have found that fluoropyrimidine therapy has poor efficacy in tumors expressing p53 mutants [112,113] and better efficacy in wild-type (wt) p53 tumors [20,114].

The p53 tumor suppressor protein is known to be involved in multiple central cellular processes, and it has been described as a mechanism of fluoropyrimidine resistance as it modulates the expression or activity of many molecules involved in these processes. Mechanistically, p53 is crucial in mediating the cellular response to DNA damage [115,116,117] and in transactivation/repression of several genes involved in the cell cycle and apoptosis [118,119,120]. TS was shown to bind p53 mRNA, which indicates a regulatory connection between these two proteins [121]. wt-p53 is more efficient in the inhibition of TYMS promoter activity than mutant p53-transfected cells, but no specific sequence in the TYMS promoter region could be assigned to this inhibition [121]. A specific interaction between p53 and TS is supported by the observation that in patients with wt-p53, a significantly lower amount of TS mRNA was detected compared to patients with mutated p53 [121]. Moreover, a strong link between TS, p53 activity and UNG2 levels was recently demonstrated [122]. Additionally, it is well known that p53 regulates DNA excision repair pathways, including BER [123,124,125,126]. After the removal of the uracil mediated by UNG2, p53 interacts with the endonuclease AP and stimulates its activity. Then, the damaged nucleotide is replaced by repair polymerases, and the remaining nick is sealed by DNA ligases. Interestingly, Yan et al. demonstrated that UNG depletion resensitizes p53-mutant and p53-deficient cancer cells to 5-FU, suggesting that in these cells, UNG is an attractive therapeutic target to enhance the response to TS inhibitors but not in wt-p53 cells, where the apoptosis pathway induced by 5-FU is independent of UNG status [114].

Notably, it was reported that TP expression is also significantly higher in colorectal carcinomas expressing p53 (mutated p53); although other factors, such as cytokines and growth factors, regulate TP, the role of p53 cannot be excluded [127].

Finally, p53 plays an important role in controlling pyrimidine catabolism by repressing the expression of DPD. Indeed, the loss of functional p53 signaling, a typical late-stage event in colorectal cancer, was accompanied by a higher expression of DPYD in advanced-stage colorectal tumor patients, which predicts poor disease-free survival [128].

In summary, the mechanisms described above explain why the overall response rate of advanced colorectal cancer to 5-FU alone is still only 10–15%. Combining 5-FU with other antitumor drugs has merely improved the response rates to 40–50% [129]. Thus, based on the biological mechanisms by which tumors acquire resistance to fluoropyrimidines, new therapeutic combination strategies are urgently needed to overcome drug resistance.

## 5. The Role of HDACis in Combination with Fluoropyrimidine-Based Therapy

The TYMS gene has been demonstrated to be one of the most prominent gene downregulated by HDACi treatment, suggesting the association of this class of drugs with fluoropyrimidines. Lee et al. reported that the HDACi trichostatin A can reverse 5-FU resistance in human cancer cells, including those of colon cancer, by downregulating TS. Trichostatin A and cycloheximide cotreatment restored TS mRNA expression, suggesting that this mechanism is regulated by an unknown transcriptional repressor [130]. Moreover, it was found that the TS protein interacted with the heat shock protein (Hsp) complex and that trichostatin A treatment induced chaperonic Hsp90 acetylation and subsequently enhanced Hsp70 binding to TS, leading to proteasomal degradation of the TS protein [131].

We and others have previously demonstrated the synergistic antitumor effects of different HDACis in combination with fluoropyrimidines in different tumors, such as breast, colorectal [19,20,21,132,133] and head and neck squamous cell carcinomas (HNSCCs) [134]. The synergistic antitumor effect of vorinostat with 5-FU was also observed in CRC cells selected for resistance to 5-FU (HT29FU cells) and in cells carrying amplification of the TS gene (H630-R10 cells), suggesting a potential mechanism by which vorinostat may overcome resistance to 5-FU as well as to another TS inhibitor, raltitrexed (RTX). In CRC cells, the antitumor activity of vorinostat is paralleled by a downregulation of TS protein expression, independent of p53 status [20].

Recently, the role of UNG2 and TS in the synergistic action of HDACis combined with pemetrexed and RTX in cells lacking p53 activity was demonstrated. Different HDACis, such as vorinostat (SAHA), entinostat (MS275), valproic acid (VPA), and sodium butyrate, induce hyperacetylation of UNG2, facilitating its interaction with a ubiquitin ligase, which thereby results in the degradation of UNG2 by the proteasome and the promotion of apoptosis [114,135,136].

Furthermore, HDACis decreased mutant p53 while stabilizing wt-p53 protein expression, a critical mechanism given that p53 status plays a critical role in anticancer drug responses, including responses to fluoropyrimidines, and patient prognosis. Alzoubi et al. investigated the role of HDAC2 in drug resistance and its impact on CRC cell lines with varied p53 mutation states (wt, null or mutated), demonstrating that increased expression of HDAC2 correlated with drug resistance, and depletion by shRNA or inhibition by HDACi sensitized the multidrug resistance of the p53 mutated HT-29 cell line to chemotherapeutic drugs such as 5-FU and oxaliplatin [137].

Interestingly, we have recently demonstrated that the synergistic interaction between HDACis and 5-FU was dependent on both the downregulation of TS and on the upregulation of TP, both induced by HDACis [19,134]. We showed that simultaneous exposure to equitoxic doses of the HDACi vorinostat plus 5-FU/CDDP produced strong synergistic antiproliferative and proapoptotic effects related to cell cycle perturbation and DNA damage induction in squamous cancer cell models. Mechanistically, vorinostat reverted 5-FU/CDDP-induced EGFR phosphorylation and nuclear translocation, leading to the impairment of nuclear EGFR noncanonical induction of genes such as TS and cyclin D1 as well as to the induction of TP [134]. We also showed that HDAC3 appears to be the HDAC isoform principally involved in TP upregulation [19]. These observations could be clinically relevant since HDAC3 has recently emerged as a critical anticancer target [138,139,140], and more selective HDAC3 inhibitors may have more favorable side-effect profiles than class-I or nonselective HDACis. In line with these observations, we recently showed that the combination of the HDACi VPA plus capecitabine synergizes with radiotherapy (RT) in CRC models, also confirming the modulation of both TS and TP protein levels by VPA even in the presence of RT [19,141] (Figure 4).

TP expression can also be regulated by other epigenetic drugs, probably due to the cross-talk between different epigenetic regulatory mechanisms. Guarcello et al. showed that the methylation of the CpG sites on the TYMP promoter region mediated the suppression of TP expression [142]. Conversely, HDACis, including tricostatin A, suberoylanilide hydroxamic acid and VPA, increased TP at both the mRNA and protein levels [132,143]. The mechanism through which HDACis mediate TP induction is not well defined. Puppin et al. showed that it does not occur through the known inducer cytokine TNFα [143]. In addition, Terranova-Barberio et al. demonstrated that specific inhibition of HDAC3 upregulated TP expression at both the transcriptional and protein levels in breast cancer cells but not in a nontumorigenic breast cell line [132].

## 6. DNA Methylation and Histone Deacetylation

DNMT inhibitors and HDAC inhibitors synergistically affect chromatin states and lead to a more pronounced re-expression of epigenetically silenced tumor suppressor genes and cell cycle regulators [81]. For instance, inactivation of TP was associated with hypermethylation of CpG dinucleotides located in the SP1-binding sites on the TP promoter. Nishizawa Y et al. demonstrated that a 5-aza-2-deoxycytidine (5-aza-CdR) demethylase inhibitor potentiated the anticancer activity of 5-FU by inducing TP expression in lung cancer cells, human epidermoid carcinoma cells, human breast ductal carcinoma cells, and human uterine cervical carcinoma cells [144]. Moreover, overexpression of molecular lysine-specific histone demethylase 1 (LSD1), a histone-modifying enzyme responsible for demethylating histone H3 lysine 4 (H3K4) and histone H3 lysine 9 (H3K9), correlated with 5-FU resistance in human CRC specimens, and the small molecule LSD1 inhibitor ZY0511 combined with 5-FU synergistically suppressed CRC tumor proliferation and metastasis, both in vitro and in vivo, by targeting Wnt/β-catenin signaling and pyrimidine metabolic pathways [145].

There is no evidence from the literature of the direct impact of HDACis on DPD expression; however, we can hypothesize that HDAC inhibitors acting on p53 can indirectly regulate DPD. It has also been reported that DPD expression can be suppressed by H3K27 trimethylation (H3K27me3) at the DPD promoter, leading to increased resistance to 5-FU [146].

Bromodomain and extraterminal motif (BET) proteins, which accumulate on transcriptionally active regulatory elements and read the state of acetylated chromatin, are important for the promotion of gene transcription, including that of many well-known oncogenes. Indeed, BET inhibitors and HDACis share many targets affecting similar cellular processes, which suggests that the inhibition of both of these classes of proteins could be an interesting strategy for improving the effectiveness of standard cancer therapy [147].

A recent study demonstrated that the combination of 5-FU with bromosporine, a novel BET inhibitor, induced cell cycle arrest and apoptosis in CRC cells and mouse models and that inhibition by bromosporine or knockdown of the BET protein BRD4, which is upregulated in HCT116 5-FU-resistant cells, might overcome 5-FU resistance [148]. In addition, Tan et al. demonstrated that a BET inhibitor markedly improved the therapeutic efficacy of anticancer agents, including 5-FU or oxaliplatin, in CRC cells by inducing death receptor 5 (DR5). This mechanism likely involves both p53-dependent and p53-independent mechanisms, leading to stronger apoptotic signaling via both the intrinsic and extrinsic apoptotic pathways [149].

Recent data have shown that BET and HDAC inhibitors exert a synergistic effect on cellular processes in cancer cells; thus, dual BET/HDAC inhibitors have been designed, and preclinical studies are ongoing [150,151]. Unfortunately, the association of BET/HDAC inhibitors or dual inhibitors with fluoropyrimidines has not been tested in clinical studies. No ongoing clinical trials are testing the combination of BET inhibitors plus fluoropyrimidines, although this combination might produce interesting results.

Regardless, it is clear that the pathways regulating fluoropyrimidine metabolism and efficacy are subject to epigenetic modifications that can influence the efficacy of the treatment, suggesting that epigenetic modifiers are attractive cancer therapeutic targets to be exploited in combination therapy. According to the recent preclinical literature discussed above, of all epigenetic drugs, HDACis seem to be the most efficacious in combination with fluoropyrimidines, and indeed, several clinical studies have explored the potential of this therapeutic strategy.

## 7. Clinical Trials

A large number of clinical trials have been conducted or are ongoing with HDACis in cancer treatment. Approximately 300 studies have been completed, and 70 are still recruiting patients, many of whom have hematological malignancies. Interestingly, although some trials have suggested that HDACis are potentially promising anticancer agents, only 6 ongoing clinical trials are in phase III, indicating that HDACis require further research in both preclinical and clinical studies before their use is established in clinical practice. In this section, we discuss published and ongoing studies that tested the association of HDACis and fluoropyrimidine-based regimens in solid tumors, where fluoropyrimidines are still the backbone of treatment despite the introduction of new anticancer treatments, such as targeted therapy and immunotherapy (Table 1).

Both 5-FU and capecitabine are currently approved in the USA and in Europe for the treatment of colorectal, esophageal, gastric and breast cancer and have been shown to be active, both alone and in combination with other chemotherapeutic agents, in a variety of other tumors, including head and neck and pancreatic cancers.

Despite many preclinical studies demonstrating the potentially synergistic antitumor effects of HDACis and fluoropyrimidines, few clinical studies have been performed, and the data reported did not show any evidence of clinical benefits from this association.

Two early phase I clinical studies testing HDACis in combination with fluoropyrimidines in solid tumors have been completed. The first phase I trial studied the feasibility and the maximum tolerated dose (MTD) of vorinostat (VOR), once or twice daily, and capecitabine, twice daily on days 1–14. This combination was administered every 21 days for at least 6 cycles, in the absence of disease progression or unacceptable toxicity, to treat patients with unresectable or metastatic solid tumors. Three dose levels were evaluated (VOR (mg/day)/CAP (mg/bid)): 300/750, 300/1000 and 400/1000. Although 3 DLTs occurred (1 at dose level 1 and two at dose level 3), the recommended doses were determined to be VOR 300 mg/day and CAP 1000 mg/bid (ClinicalTrials.gov Identifier: NCT00121277) [162].

The second trial studied the feasibility and MTD of the combination of belinostat (PXD101) and 5-fluorouracil in patients with advanced solid tumors. Patients received dose escalation of both belinostat (300, 600, or 1000 mg/m^2^ IV for 5 days every 21 days) and 5-fluorouracil (250, 500, 750, or 1000 mg/m^2^/day) administered as a continuous 96 h infusion, with belinostat starting on day 2 of cycle 2 onward. Five doses of PXD101/5-FU (mg/m^2^/day) were evaluated: 300/250, 600/250, 1000/250, 1000/500, and 1000/1000, and the combination of PXD101/5-FU was well tolerated up to a dose of 1000/500 mg/m^2^/day (ClinicalTrials.gov Identifier: NCT00413322) [163].

In colon cancer, combinations of fluoropyrimidines, either 5-FU or capecitabine, in association with oxaliplatin or irinotecan constitute the basis for the treatment of patients with metastatic disease.

A phase I study evaluated vorinostat administered for 7 consecutive days every 14 days in combination with a standard modified FOLFOX6 regimen administered at a fixed dose on days 4 and 5 of vorinostat administration [152]. Twenty-one patients were enrolled, and MTD was established at 300 mg PO BID vorinostat in combination with a standard dose of FOLFOX, resulting in fatigue and dose-limiting toxicity. This study also included a pharmacokinetic evaluation of vorinostat, 5-FU, and oxaliplatin. The schedule of vorinostat administration used in this trial resulted in inadequate modulation of thymidylate synthase (TS) expression, which the authors suggested probably explained the lack of significant clinical activity of the combination and speculated that a shorter intermittent dosing may allow for a higher dose administration/day and a suitable blood concentration of vorinostat.

A phase I/II study published by Wilson et al. in 2010 failed to establish the MTD of vorinostat, administered at a dose of 400 mg daily for 14 or 7 days every 2 weeks, in association with 5-FU/LV. Most likely, the fact that all 10 patients enrolled were heavily pretreated contributed to the overall toxicity in the study [153]. Consistent with the study of Fakih et al. [152], in this study, the treatment was unable to produce consistent decreases in intratumoral TS expression, despite the biological activity of vorinostat, which was confirmed by the evaluation of histone acetylation on PBMCs, again pointing to the need for an alternate vorinostat dose schedule [153].

Indeed, the overall toxicity observed in the previous studies was not observed in a randomized phase I/II study evaluating an intermittent dose vorinostat, at two different high dosages (43 patients were treated with 800 mg/day and 15 patients were treated with 1400 mg/day once a day for three days, every 2 weeks), in combination with 5-FU/LV (ClinicalTrials.gov Identifier: NCT00942266) in chemorefractory metastatic colorectal cancer patients [154]. Overall, these results suggested that continuous dosing of vorinostat was crucial to its toxic effects. Regardless, in both arms of the last phase II study, the authors did not see significant signs of activity of the combination. In particular, the group treated with 1400 mg/day vorinostat closed after the first stage due to lack of activity. Nevertheless, only 1 partial response was observed among of the 58 patients enrolled, and the authors reported an interesting median OS of 6.5 months. The potential explanations for the negative outcomes of these studies vary. First, the selection of patients and the choice of treatments should be considered. In both phase II studies, the patients enrolled were refractory to 5-FU, and in the study published by Wilson et al., all patients were eligible even if they had a high level of intratumoral TS expression, which has been described as a clear mechanism of resistance to 5-FU. Although preclinical data reported that the HDACi vorinostat was able to overcome 5-FU resistance in a TS overexpression cell line as well as in a subline selected for adaptation to 5-FU [20], it is well known that preclinical data do not always translate into clinical results since the response of patients to drug treatments depends on many more variables compared to in vitro cell lines. A more appropriate clinical investigation would have focused on non5-FU-resistant colorectal cancer patients and used different HDACis to reduce toxicity. Based on these ideas, two clinical trials are ongoing at our Institute evaluating the association of HDACi VPA and fluoropyrimidine-based regimens in locally advanced rectal cancer (LARC) and in first-line treatment of metastatic colorectal cancers [28,155]. A phase I/II clinical trial was designed to demonstrate the feasibility and activity of VPA (administration based on a titration strategy described below; up to 500 mg three times a day) in association with preoperative treatment with short course radiotherapy (SCRT), a very convenient modality of RT, in combination with capecitabine in LARC patients with a low–moderate risk of relapse (ClinicalTrials.gov Identifier: NCT01898104). Preliminary results from phase I demonstrated that the addition of capecitabine to preoperative SCRT +/− VPA was feasible, and 825 mg/m^2^/bid was the recommended dose that will be used in an ongoing phase-2 trial [164]. Interestingly, VPA treatment did not have a predefined dose, but a titration strategy was applied in each patient to achieve a serum concentration between 50 and 100 μg/mL. This target serum level was the recommended value for the treatment of epilepsy. Specifically, VPA was administered orally starting at day −14, with 500 mg slow releasing tablets provided in the evening. Thereafter, the dose was also increased using 300 mg tablets until it reached 1500 mg on day −1; thus, VPA was administered orally on days −14 to 21, in association with SCRT or capecitabine, and was well tolerated.

Interestingly, the second clinical trial ongoing at our Institute (a randomized, open-label, two-arm, phase II study) explores the addition of VPA administered with the same scheme as the previous study to first-line bevacizumab/oxaliplatin/fluoropyrimidine regimens (mFOLFOX-6/mOXXEL) in RAS-mutated metastatic colorectal cancer patients (Clinical trial information: NCT04310176) [155]. The large number of correlative studies planned in both clinical trials may provide new insight into the mechanism of interaction between HDACis, and VPA in particular, and fluoropyrimidines.

In breast cancer patients, four different HDAC inhibitors have been tested in association with capecitabine to identify the MTD and/or DLT. Peacock N et al. presented a Phase I study of panobinostat (LBH589) with capecitabine and with or without lapatinib administered to breast cancer patients with pretreated advanced tumors for which capecitabine was clinically appropriate at the ASCO annual meeting in 2010 [156].

The study was designed with three objectives: to establish the MTD and DLTs of panobinostat in combination with capecitabine (Part 1); to assess the safety and tolerability of panobinostat in combination with lapatinib (Part 2); and to evaluate the tolerability and effectiveness of the triple combination of panobinostat, capecitabine, and lapatinib in women with metastatic breast cancer (Part 3). The administration of 30 mg oral panobinostat twice weekly was feasible and safe in association with two different dosages of capecitabine [825 (4 patients) and 1000 (11 patients) mg/m^2^] BID for 14 out of 21 days. The study completed Part 1 with 15 patients and reported that the combination of panobinostat and capecitabine was well tolerated at the recommended doses of 30 mg twice weekly and 1000 mg/m^2^ BID for 14 days every 21 days, respectively. However, the dosing schedule for panobinostat was changed to 20 mg three times weekly for Parts 2 and 3 of the study. Five patients have been enrolled in Part 2, which evaluates the association of panobinostat and lapatinib. Only one patient with metastatic breast cancer has had an objective response and 27% had stable disease.

Poor (moderate) activity was also reported for the combination of vorinostat and capecitabine, which were tested in a phase I study in 23 advanced breast cancer patients [165]. Only 14 patients were evaluable for clinical response, but no objective responses were seen; 3 patients had stable disease lasting more than 6 months (Clinical trial information: NCT00719875).

The third phase I study of the HDAC inhibitor entinostat in association with capecitabine started recently and is ongoing (ClinicalTrials.gov Identifier: NCT03473639). The purpose of the study is to evaluate the safety and feasibility of the combination of entinostat and capecitabine in patients with metastatic breast cancer or high-risk breast cancer after neoadjuvant therapy.

VPA has also been tested in a phase I/II study in combination with FEC100 (epirubicin 100 mg/m^2^ with 5-fluorouracil 500 mg/m^2^ and cyclophosphamide 500 mg/m^2^) in 44 solid cancer patients (41 evaluable for response) with a disease-specific cohort expansion of 15 patients (14 evaluable for response) pretreated locally advanced or metastatic breast cancer patients receiving 120 mg/kg/day valproic acid followed by FEC100. The administration of VPA was a loading dose (15, 30, 45, 60, 75 90, 100, 120, 140, and 160 mg/kg/day) followed by 5 oral doses in 2 divided doses (7.5, 15, 22.5, 30, 37.5, 45, 50, 60, 70, and 80 mg/kg) given every 12 h starting 4 h after the loading dose. Interestingly, partial responses were seen in 9 of 41 patients (22%). In the expansion cohort, objective responses were seen in 9 of 14 evaluable patients (64%) at dose expansion with a median number of 6 administered cycles. Somnolence was the predominant toxicity associated with VPA [157] (ClinicalTrials.gov Identifier: NCT00246103).

Finally, in phase II of the study, VPA (60 mg/kg BID) was tested in combination with FEC100 in patients with locally advanced or primary metastatic breast cancer. The study was closed prematurely due to a lack of efficacy, enrolling only 6 of the 55 estimated patients (ClinicalTrials.gov Identifier: NCT01010854).

Only a few studies have been conducted to evaluate the association of HDACis and fluoropyrimidine-based regimens in gastric cancer. In a phase I trial, 23 patients were enrolled to study the side effects and MDT of vorinostat when administered with irinotecan, fluorouracil, and leucovorin (FOLFIRI) in patients with advanced upper gastrointestinal cancers, including esophageal, gastric and liver cancers (NCT00537121). Ten patients were treated with 3 dose levels of vorinostat (2 at 200 mg, 5 at 300 mg, and 3 at 400 mg). No DLT was noted at any dose level, and of the 8 patients evaluable for response, 2 patients experienced a partial response, and 5 patients had stable disease [166]. The study, conducted by Roswell Park Cancer Institute, was completed and closed in 2013, but unfortunately, the data were not published.

In a Korean phase I/II study in advanced gastric cancer, vorinostat was associated with the standard combination of capecitabine and cisplatin. The dose escalation of each drug was tested starting from vorinostat 300 mg/day, cisplatin 60 mg/m^2^, and capecitabine 1600 mg/m^2^/day up to vorinostat 400 mg/day, cisplatin 80 mg/m^2^, and capecitabine 2000 mg/m^2^/day with a standard 3 + 3 method.

In phase I, a total of 30 patients with unresectable or metastatic gastric adenocarcinoma were enrolled, and the recommended doses for further development were vorinostat 400 mg/day, cisplatin 60 mg/m^2^, and capecitabine 2000 mg/m^2^/day every three weeks (ClinicalTrials.gov Identifier: NCT01045538). Histone H3 acetylation in PBMCs was monitored to identify a possible biomarker that could predict efficacy and toxicity in patients treated with vorinostat. Interestingly, a significant correlation between the levels of H3 acetylation and the dose of vorinostat was observed, and a greater increase in H3 acetylation after vorinostat administration was associated with lower baseline H3 acetylation levels [158].

In phase II of the study, 45 patients with HER2-negative unresectable or metastatic gastric cancer were enrolled. The results showed that vorinostat was ineffective in enhancing the efficacy of the capecitabine and cisplatin combination in these patients and that the addition of vorinostat induced more adverse events in comparison with the previous history of fluoropyrimidine–platinum doublet regimens. Biomarker analysis revealed that high plasma acetyl-H3 and p21 levels were significantly associated with poor OS, suggesting their possible role as predictive markers of efficacy [159].

Slightly more encouraging data were observed in pancreatic cancer patients, one of the deadliest cancers in which a more effective therapeutic approach is needed.

Iwahashi S et al. conducted a phase I/II clinical trial to examine the safety and efficacy of a VPA and tegafur combination in twelve patients with advanced pancreatobiliary tract cancers in whom curative surgery was not feasible [160]. Patients received a daily dose of 80 mg/m^2^ oral tegafur for 28 days, followed by a 14 days recovery period when they received VPA orally twice daily at a total dose of 15 mg/kg/day. Although grade 3/4 adverse events, including anemia and platelet depletion, were observed, the results were somewhat intriguing. Anemia and platelet depletion are two toxicities commonly observed when treated with tegafur alone. Although in this trial the partial response rate was lower than that reported for monotherapy with tegafur, the disease control rate (PR and SD) of the tegafur and VPA combination compared favorably with the disease control rate reported for tegafur alone (91.7% vs. 63.3%) [167]. Moreover, significant increases in blood concentrations of VPA were confirmed 2 and 4 weeks after VPA administration. In a phase I study, twenty-one patients with nonmetastatic pancreatic cancers received escalating doses of vorinostat (100–400 mg daily) in association with capecitabine and radiotherapy. Vorinostat was administered at every assigned daily dose level (100, 200, 300, or 400 mg) during radiotherapy (30 Gy in 10 fractions) and for the following two weeks after radiation was completed, while 1000 mg q12 capecitabine was administered on the days of radiation. The MTD of vorinostat was 400 mg, and although DLT occurred in one patient at each dose level, the combination of vorinostat with capecitabine and RT was feasible and well tolerated. Interestingly, 11 out 12 borderline resectable patients underwent exploratory surgery, and there were four R0 resections (microscopic margins negative for tumor) and one R1 resection (microscopic margins positive for tumor). Finally, an encouraging median OS of 1.1 years was reported (ClinicalTrials.gov Identifier: NCT00983268) [161]. A phase I/II study of longer duration, which started in 2009 and closed in 2017, tested vorinostat in association with 5-FU and radiotherapy in patients with locally advanced pancreatic cancer (ClinicalTrials.gov Identifier: NCT00948688) with the aim of finding the MDT of vorinostat in this combination. This trial enrolled only 10 of the 50 planned patients, and phase II of the study was not performed. Two different doses of vorinostat (200 mg or 100 mg orally, days 1–7, weeks 1–6) were combined with 5-FU (225 mg/m2/day IV; days 1–5, weeks 1–6 until completion of radiation therapy) and radiotherapy (180 cGy daily Monday–Friday; 28 days of treatment), but the MTD was not determined due to premature closure of the study. Finally, a recently started phase Ib study aims to determine the recommended dose of entinostat in combination with the standard FOLFOX chemotherapy regimen in metastatic pancreatic cancer patients, which will be evaluated in a subsequent phase II study. The secondary objectives are to assess the safety, tolerability and efficacy of the combination. The study is a modified 3 + 3 dose-escalation design. All patients will receive the same dose of FOLFOX once every 2 weeks and various doses of entinostat (2, 3, 4 or 5 mg) on days 1, 8, 15, and 22 of the 28 day cycles (ClinicalTrials.gov Identifier: NCT03760614).

The combination of vorinostat plus capecitabine was also assessed in a nonrandomized two-stage open-label study of patients with recurrent and/or metastatic squamous cell carcinoma of the head and neck (HNSCC) and recurrent and/or metastatic nasopharyngeal carcinoma (NPC) (Stage I), followed by a randomized study of patients with NPC (Stage II) (ClinicalTrials.gov Identifier: NCT01267240). Twenty-five patients were enrolled to receive capecitabine BID and vorinostat daily on days 1–14. Every 21 days, the treatment was repeated in the absence of disease progression or unacceptable toxicity. The response was assessed in 16 of the 25 enrolled patients, but clinical activity was inconsistent.

## 8. Conclusions

At present, the identification of drugs that can overcome fluoropyrimidine resistance remains a clinical priority. In this manuscript, we reviewed multiple varying mechanisms responsible for resistance to fluoropyrimidines, highlighting many novel potential therapeutic targets. Moreover, we demonstrated that epigenetic agents such as HDACis can reverse fluoropyrimidine resistance by targeting specific genes or proteins. Although preclinical evidence showed a strong synergistic interaction between HDACis and fluoropyrimidines in different cancer models, the data from clinical studies did not support the preclinical observations. It is well known that, despite the introduction of increasingly complex in vitro and in vivo preclinical models, these models cannot recapitulate human complexity. Moreover, the analysis of the clinical studies revealed that the majority of these studies lacked a clear and mechanistic approach.

## Figures and Tables

**Figure 1 cancers-14-00695-f001:**
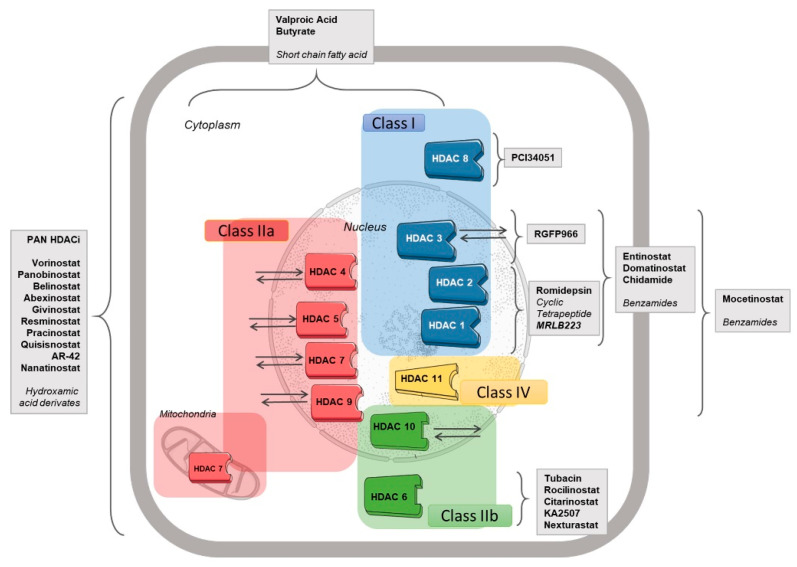
Classification of HDAC, their cellular localization and their inhibitors (HDACis). According to their structure and function, HDAC proteins are grouped into four classes. Class III deacetylases, sirtuins proteins, are not depicted here.

**Figure 2 cancers-14-00695-f002:**
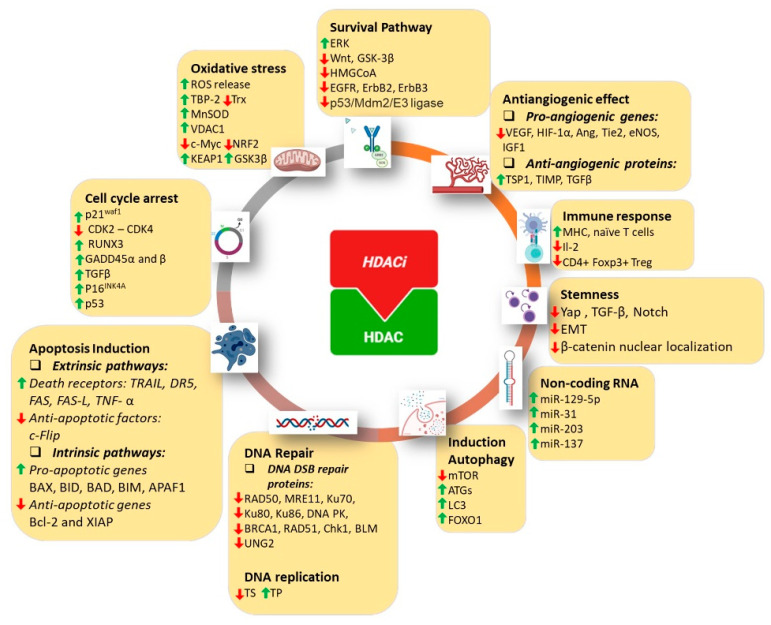
Mechanisms of the anticancer effects of HDAC inhibitors. HDAC inhibitors induced a pleiotropic effect on cancer cells, including modulation of survival pathways, angiogenesis, immune response, stemness, noncoding RNA, autophagy, DNA repair, DNA replication, apoptosis, cell cycle arrest, and oxidative stress.

**Figure 3 cancers-14-00695-f003:**
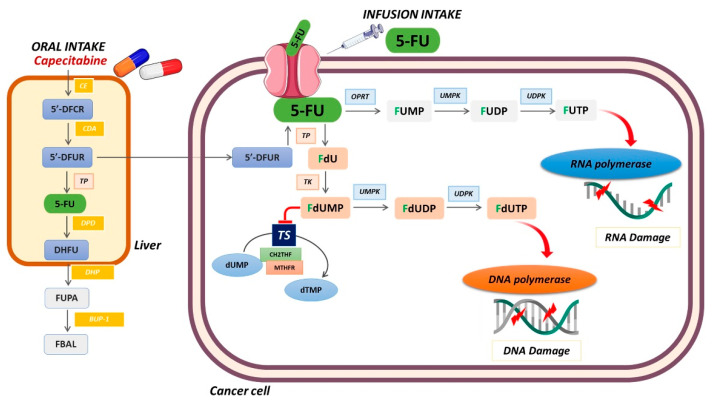
Schematic representation of fluoropyrimidine metabolism.

**Figure 4 cancers-14-00695-f004:**
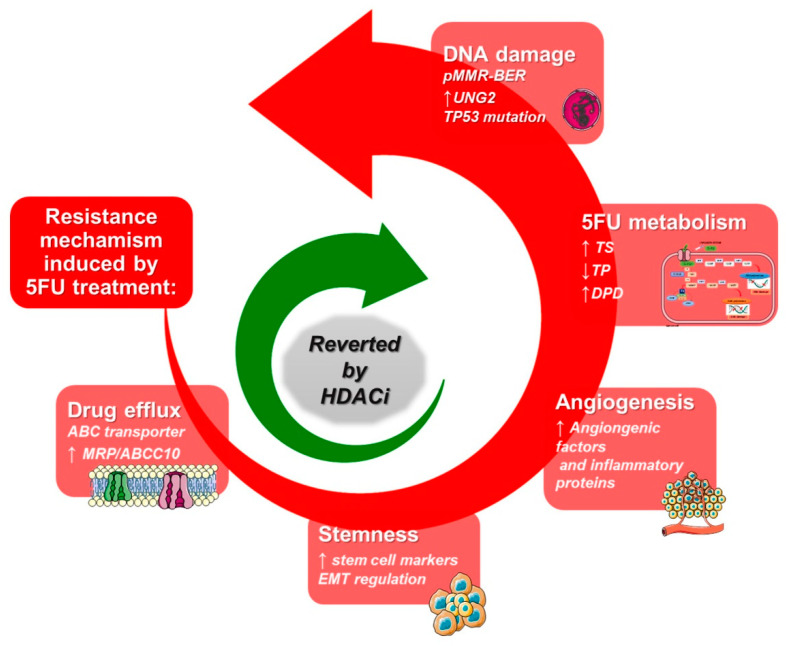
Mechanisms of overcoming fluoropyrimidines drug resistance by HDACis. Red arrow connects the mechanisms by which cancer cells acquire resistance to fluoropyrimidines, whereas green arrow indicates that the treatment with HDACis could act on these mechanisms reverting the resistance and sensitizing cancer cells to chemotherapy treatment.

**Table 1 cancers-14-00695-t001:** Overview of clinical trials evaluating the combination of HDAC inhibitors with fluoropyrimidines in solid tumors.

	Trial	Phase	Setting	Regimen	Status	Ref
**Multiorgan**	Vorinostat and Capecitabine in Treating Patients With Metastatic or Unresectable Solid Tumors	Phase I	Unresectable or metastatic solid tumors	Vorinostat + capecitabine		ClinicalTrials.gov: NCT00121277
	Study of PXD101 Alone and in Combination With 5-Fluorouracil (5-FU) in Patients With Advanced Solid Tumors	Phase I		Belinostat (PXD101) + 5-fluorouracil		ClinicalTrials.gov: NCT00413322
**Colorectal cancer**	A Phase I, Pharmacokinetic, and Pharmacodynamic Study of Vorinostat in Combination with 5-Fluorouracil, Leucovorin, and Oxaliplatin in Patients with Refractory Colorectal	Phase I	Refractory colorectal	Vorinostat and 5-fluorouracil + leucovorin + oxaliplatin	Published	[152]
	A phase I/II trial of vorinostat in combination with 5-fluorouracil in patients with metastatic colorectal cancer who previously failed 5-FU-based chemotherapy	Phase I/II	Metastatic colorectal who had failed all standard therapeutic options	Vorinostat + 5-FU/LV	Published	[153]
	Vorinostat, Fluorouracil, and Leucovorin Calcium in Treating Patients With Metastatic Colorectal Cancer That Has Not Responded to Previous Treatment	Phase II	Adenocarcinoma of the colon and rectum Recurrent colon cancer and rectal cancer Stage IV colon and rectal cancer	Vorinostat + 5-FU/LV	Published	ClinicalTrials.gov: NCT00942266. [154]
	Preoperative Valproic Acid and Radiation Therapy for Rectal Cancer	Phase I/II	Rectal cancer	Preoperative radiation therapy + valproic acid + capecitabine	Recruiting	ClinicalTrials.gov: NCT01898104 [28]
	Valproic Acid in Combination With Bevacizumab and Oxaliplatin/Fluoropyrimidine Regimens in Patients With Ras-mutated Metastatic Colorectal Cancer	Phase II	Ras-mutated metastatic colorectal cancer	Bevacizumab + mFOLFOX6 or mOXXEL regimen + valproic acid	Recruiting	ClinicalTrials.gov: NCT04310176 [155]
**Breast**	LBH589 in Combination With Capecitabine Plus/Minus (±) Lapatinib in Breast Cancer Patients	Phase I	Refractory and advanced breast cancer sensitive to 5-fluorouracil	Panobinostat + capecitabine + lapatinib	Completed	ClinicalTrials.gov: NCT00632489 [156]
	HDAC Inhibitor Vorinostat (SAHA) With Capecitabine (Xeloda) Using a New Weekly Dose Regimen for Advanced Breast Cancer	Phase I	Advanced breast cancer	Vorinostat + capecitabine	Completed	ClinicalTrials.gov: NCT00719875
	A Pilot Study of the Combination of Entinostat With Capecitabine in High Risk Breast Cancer After Neo-adjuvant Therapy	Phase I	Metastatic breast cancer	Entinostat + capecitabine	Recruiting	ClinicalTrials.gov: NCT03473639
	Phase I Trial of Valproic Acid and Epirubicin in Solid Tumor Malignancies	Phase I/II	Neoplasms, advanced (breast)	Valproic acid + FEC (epirubicin, 5-fluorouracil; cyclophosphamide)	Completed	ClinicalTrials.gov: NCT00246103 [157]
	Valproic Acid in Combination With FEC100 for Primary Therapy in Patients With Breast Cancer	Phase II	Breast cancer	VPA + FEC100	Terminated	ClinicalTrials.gov: NCT01010854
**Gastric**	Vorinostat, Irinotecan, Fluorouracil, and Leucovorin in Treating Patients With Advanced Upper Gastrointestinal Cancer	Phase I	Esophageal; gastric; liver cancer	Vorinostat + 5-fluorouracil + irinotecan hydrochloride + leucovorin calcium	Completed	ClinicalTrials.gov: NCT00537121
	Study of Vorinostat Plus Capecitabine (X) and Cisplatin (P) for 1st Line Treatment of Metastatic or Recurrent Gastric Cancer: Zolinza+XP	Phase I/II	Gastric cancer	Vorinostat + capecitabine + cisplatin	Completed	ClinicalTrials.gov: NCT01045538 [158,159]
**Pancreas**	Effects of Valproic Acid in Combination with S-1 on Advanced Pancreatobiliary Tract Cancers: Clinical Study Phases I/II	Phase I/II	Advanced pancreatobiliary tract cancers	VPA + tegafur	Published	[160]
	Capecitabine, Vorinostat, and Radiation Therapy in Treating Patients With Nonmetastatic Pancreatic Cancer	Phase I	Nonmetastatic pancreatic cancers	Vorinostat + capecitabine + radiotherapy + surgery	Completed	ClinicalTrials.gov: NCT00983268 [161]
	Vorinostat With XRT and 5-FU for Locally Advanced Adenocarcinoma of the Pancreas	Phase I/II	Pancreas adenocarcinoma	Vorinostat + radiation therapy + 5-FU	Terminated	ClinicalTrials.gov: NCT00948688
	A Study of Entinostat and FOLFOX in Subjects With Pancreatic Adenocarcinoma	Phase I	Pancreas cancer	Entinostat + FOLFOX regimen	Not yet recruiting	ClinicalTrials.gov: NCT03760614
**Head and neck**	Capecitabine and Vorinostat in Treating Patients With Recurrent and/or Metastatic Head and Neck Cancer	Phase II	HNSCC	Vorinostat + capecitabine	Terminated	ClinicalTrials.gov: NCT01267240

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
