# Peer review of "Epigenetic Approaches to Overcome Fluoropyrimidines Resistance in Solid Tumors"

_cancers, 2022, doi:10.3390/cancers14030695_

Round 1
Reviewer 1 Report
In their paper entitled: "Epigenetic approaches to overcome fluoropyrimidines resistance in solid tumors", the authors provide an overview of fluoropyrimidines, their effect in cancer treatment, the mechanism of cancer cell resistance and the role of histone deacetylase in combination therapy with fluoropyrimidines. The work ranges from basic molecular information to examples from clinical trials. This work is well written, consistent and complex. I have only a few minor comments:
Figure 1: Citoplasm should be cytoplasm.
Figure 2: Explain the arrows (green, red) in the legend of this figure.
Author Response
We would like to thank for the positive comments and we have revised the manuscript as suggested:
1) we have changed “citoplasm” in “cytoplasm” in Figure 1
2) we have added a brief description of the Figure 2 explaining the meaning of the arrows.
Reviewer 2 Report
The review “Epigenetic approaches to overcome fluoropyrimidines resistance in solid tumors“ by Laura Grumetti et al. highlights the mechanisms by which epigenetic modifiers like HDAC inhibitors can overcome the resistance of fluoropyrimidines in tumor therapy and may reduce their toxicity. The authors are focused on solid tumors in their descriptions.
The topic of this review is highly relevant and important. Current research strives to develop specific treatment schemes to reduce unwanted side effects, toxicity, and drug resistance in cancer therapy.
The review is well-written, well-structured, and very informative. It is comprehensive and includes the biochemical mechanisms of resistance to fluoropyrimidines and the role of HDAC inhibitors to overcome this resistance.
Grumetti et al. are not only focusing on biochemical studies they also included actual clinical trials and tried to point out the strength and weakness of these studies.
Their conclusions are comprehensible and underlined by actual publications.
In my opinion, this is a very well-written review I would recommend to publish in the present form.
Author Response
We would like to thank for the positive comments.
Reviewer 3 Report
In this review, the authors provided a comprehensive review of the mechanisms by which histone deacetylase inhibitors (HDACis) prevent/overcome the resistance and/or enhance the therapeutic efficacy of fluoropirymidines, including several preclinical and clinical studies. The manuscript is very well written, a good overview and updated studies. I would recommend a minor revision to this review prior to publication.
- In line 61, use “histone acetyltransferases” instead of “histone acetylase”.
- SP1 or Sp1. It should be consistent in the manuscript.
Author Response
We would like to thank for the positive comments and we have revised the manuscript as suggested:
1) we have replaced “histone acetylase” with “histone acetyltransferases” in line 61 of the manuscript
2) we have changed “Sp1” in “SP1” throughout the manuscript.